# Immunity after HPV Vaccination in Patients after Sexual Initiation

**DOI:** 10.3390/vaccines10050728

**Published:** 2022-05-06

**Authors:** Dominik Pruski, Małgorzata Łagiedo-Żelazowska, Sonja Millert-Kalińska, Jan Sikora, Robert Jach, Marcin Przybylski

**Affiliations:** 1Department of Obstetrics and Gynecology, District Public Hospital in Poznan, 60-479 Poznań, Poland; millertsonja@gmail.com (S.M.-K.); nicramp@poczta.onet.pl (M.P.); 2Gynecology Specialised Practise, 60-408 Poznań, Poland; 3Department of Immunology, Chair of Pathomorphology and Clinical Immunology, Poznan University of Medical Sciences, 60-806 Poznań, Poland; mlagiedo@ump.edu.pl (M.Ł.-Ż.); jan-sikora@wp.pl (J.S.); 4Doctoral School, Poznan University of Medical Sciences, 61-701 Poznań, Poland; 5Department of Gynecological Endocrinology, Jagiellonian University Medical College, 31-008 Cracow, Poland; jach@cm-uj.krakow.pl

**Keywords:** HPV serum antibodies, L1 HPV, 9-valent vaccination, squamous intraepithelial neoplasia

## Abstract

Vaccinations against human papillomavirus (HPV) are included in the primary prevention of precancerous intraepithelial lesions and HPV-related cancers. Despite the undeniable effectiveness of vaccination in the juvenile population, there is still little research on the effect in patients after sexual initiation. Our study aims to assess anti-HPV (L1 HPV) antibodies in healthy patients and diagnosed cervical pathology after 9-valent vaccination. We provide a prospective, ongoing 12-month, non-randomised pilot study in which 89 subjects were enrolled. We used an enzyme-linked immunosorbent assay to determine IgG class antibodies to HPV. We noted significantly higher levels of antibodies in vaccinated individuals than in the unvaccinated control group. The above work shows that vaccination against HPV might be beneficial in patients after sexual initiation as well as in those already diagnosed with HPV or SIL infection.

## 1. Introduction

Squamous intraepithelial lesion (SIL) and cervical cancer are among the most common oncological diagnoses in women globally; therefore, they constitute a significant health problem. Cervical cancer remains the fourth most frequent cancer in women worldwide [1] unless it is theoretically preventable. The most critical risk factor for the development of cervical cancer is a persistent infection caused by highly oncogenic types of human papillomavirus (HPV). Neoplastic transformation begins with integrating HPV DNA into the genome of a typical epithelial cell. This situation may occur when the circular form of HPV DNA breaks, and then chromatin shifts within the chromosomal DNA of the host cells. Vaccination against HPV prevents infections with specific HPV types and, consequently, cervical cancer development due to infection with a given type [2,3,4]. Generally, during the human immune response to HPV, B cells detect the viral antigens and exhibit them to T helper type 2 cells, promoting the production of high-affinity antibodies (IgG, IgA, and IgM) against HPV antigens by B cells. It has already been demonstrated that anti-HPV IgG might be a reliable marker for past HPV exposure [5]. Studies have shown that the median seroconversion time was about 8.3–11.8 months. These data suggest that the development of IgG antibodies at a detectable level after a natural infection can be a slow process, and it does not necessarily occur in every woman. Following a human papillomavirus (HPV) vaccine, type 16 virus-like particles (VLPs), according to the authors, appear within 8.3 months and remain for approximately 36 months [6]. Antibodies could persist for a long period of time if the initial antibody levels were high or if there was continued antigenic exposure. At the same time, IgM may be detected in acute or current exposure, typically after one month following initial immunisation, as Harro et al. claim [7]. Vaccinations against HPV are included in the primary prevention of precancerous lesions—mainly SIL and cervical cancer. The other cancers associated with HPV infections affect the genital organs (vulva, vagina, and penis), anal canal, oral cavity, and upper respiratory tract [8,9]. Vaccination against HPV significantly reduced the incidence of HPV-related lesions in New Zealand and the United States [10,11]. In the European countries, and thus in Poland, vaccination against HPV has been introduced into the vaccination calendar. Local governments organise vaccination programmes in many provinces of our country. It is recommended that both girls and boys are vaccinated before sexual initiation. After identifying an HPV infection, many patients decide to vaccinate after sexual initiation due to the fear of developing intraepithelial neoplasia of the cervix, vagina, vulva, or HPV-dependent changes in the respiratory tract. After the treatment of HPV-related lesions, such as intraepithelial neoplasia of the cervix or genital warts, some patients decide to vaccinate to develop anti-HPV antibodies that can protect against re-infection and the formation of HPV-related lesions. However, there is still very little research into post-vaccination antibody levels (VLP), so it seems to us that this is a topic worth exploring [12,13,14].

The 9-valent vaccine contains the purified proteins of nine types of HPV, namely 6, 11, 16, 18, 31, 22, 45, 52, and 58. The vaccine is usually administered according to a three-dose schedule. Studies focusing on the presence of HPV genotypes in large populations may contribute to the development of further protective vaccinations [15]. 

Considering this, we aim to assess the level of anti-HPV (L1 HPV) antibodies in healthy patients and with diagnosed cervical pathology after vaccination. The introduction of tests for the detection of anti-HPV (L1 HPV) antibodies may, in the future, facilitate the assessment of the effectiveness of vaccine programmes. Moreover, it might be helpful in the identification of patients with immune disorders in whom infection with oncogenic types of HPV persisted, resulting in intraepithelial neoplasia. Analysing specific types of immune disorders will facilitate the identification of groups of women with the highest risk of developing high-grade squamous intraepithelial lesions and, consequently, malignancy.

The following meta-analysis compares the effectiveness of the vaccine administration in the population of patients before and after sexual initiation in either healthy individuals or those with diagnosed cervical pathology.

## 2. Materials and Methods

### 2.1. Study Design

We provide a prospective, ongoing 12-month, non-randomised pilot study to assess the level of anti-HPV (L1 HPV) antibodies in healthy patients and those with diagnosed cervix pathology after the 9-valent HPV vaccine. The Bioethical Committee of the Poznan University of Medical Sciences, Poland, approved the study protocol (597/19). We obtained written consent for the study from all patients. We included patients who met the following criteria: (i) only adult women, (ii) non-pregnant subjects, postpartum, (iii) patients not treated with immunosuppressive drugs, (iv) not previously vaccinated with other HPV vaccines, (v) expressing informed and written consent to participate in the study, (vi) agreeing to the proposed surgical diagnostics in the case of indications and possible surgical treatment, (vii) had taken three doses of the 9-valent vaccination against HPV according to the 0–2–6 months scheme, and (viii) provided blood samples after at least six months from the last dose of vaccination. The exclusion criteria were: (i) refusal of possible treatment of squamous intraepithelial lesions, and (ii) failure to complete the full vaccination schedule. A total of 61 women met the above criteria.

All subjects from the study group were undergoing a verification diagnostic of abnormal Pap-smear results by punch biopsy. We examined the status of HPV infection and looked for the presence of pre-neoplastic lesions, such as low-grade squamous intraepithelial lesions (LSIL) or high-grade squamous intraepithelial lesions (HSIL). All patients with histopathologically confirmed HSIL (CIN 2, CIN 3) were treated with the LEEP conization and curettage of the cervical canal.

The control group 1 covers 20 healthy, unvaccinated patients, in whom we excluded an infection with hrHPV or squamous intraepithelial lesions confirmed through punch biopsy. Control group 2 includes eight subjects both infected with highly oncogenic types of HPV and diagnosed with pre-neoplastic lesions who decided not to receive the HPV vaccine. Figure 1 presents the process of recruiting patients for the study, and Table 1 shows the basic division into study groups.

All examination and follow-up groups are under regular oncogynaecological care. Patients diagnosed with HSIL (CIN 2, CIN 3) underwent proper treatment—the removal of the lesions according to the current recommendations of the Polish Colposcopic Society—and then subjected to close cytological and molecular control every six months.

### 2.2. Specimen Collection and Handling

Blood was drawn aseptically to the serum collection tubes (S-Monovette). The blood was collected at least six months after receiving the last vaccination dose. After that, the samples were centrifuged at 2000 rpm for 20 min. Supernatants (sera) were collected and frozen at −20 °C for further assays.

### 2.3. HPV Serological Measurements

We used an enzyme-linked immunosorbent assay to determine IgG class antibodies to human papillomavirus (Creative Diagnostics, New York, USA). The sera were diluted 1:101 into properly defined dilution tubes for the test. An ELISA microtiter plate was coated with recombinant VLP derived from HPV types 6, 11, 16, and 18. After incubation and washing, we added the diluted samples and quality control specimens to the microtiter plates along with a peroxidase-conjugated anti-human polyclonal antibody. Following incubation and washing, an enzyme substrate and chromogen were added to allow colour development. Reactions were stopped, and optical density (OD) was read at 450 and 620 nm, with the background measured at 620 nm and subtracted from the OD reading at 450 nm. A calculated formulation from the manufacturer determined the seropositive cut points. The cut points were set at 0.303 for HPV seronegative and >0.303 for HPV seropositive patients. We calculated the quantitative results of the assay as instructed in the kit insert (OD/CUT-OFF).

### 2.4. Statistical Analysis

We performed an analysis in SPSS, version 27. All tests were two-tailed, with α = 0.05. The normality of the variables was validated based on the Shapiro–Wilk test. All three groups were characterised by reporting median with quartiles 1 and 3, or mean and standard deviation for quantitative variables, or *n* value and percentage for qualitative variables. The values of variables with normal distributions were compared between the experimental and control group 1 or 2 with the Student’s *t* test. Variables without normal distributions were compared with the Mann–Whitney U test. Dependencies between the group and other variables were measured using Fisher’s exact test. Odds ratios or median differences (experimental group–control group) with 95% confidence intervals were given when the results of the analyses were significant. Median differences were calculated using the Hodges–Lehmann estimator. The correlation was calculated with Pearson’s r coefficient.

## 3. Results

As shown in Table 1, the experimental and control groups did not differ significantly in age or regarding pregnancies. Comorbidities were observed in 38% of women from the experimental group, 30% from control group 1, and 12.5% of women from control group 2. The dependency between the group and Pap-smear results was insignificant for the experimental vs. control group 2. The most common Pap-smear result in the experimental and control group 2 was LSIL, which accounted for 33% and 38%, respectively. A more significant proportion of women with positive HPV tests was found in the experimental group than in control group 1—nine times more. There was no considerable dependency between the groups and HPV test results in the case of the experimental group vs. control group 2. All women from control group 1 were histopathologically confirmed to have no pathology (NILM). An NILM result was observed in 19% of women in the experimental group, which is statistically significant. We did not find any dependency between the experimental/control group 2 and biopsy results. The most common histopathological result for women from the experimental group and control group 2 was HSIL (57% of women from both groups).

Figure 2 and Figure 3 show the graphical arrangement of the levels of antibodies in individual research groups. Antibody levels were significantly higher in the experimental group than in both control group 1 and control group 2. The antibody level divided by the cut-off value (0.303) was also significantly higher in the experimental group than both of the control groups. There were significant dependences between group and sample being reactive (*p* < 0.001 for both analysis—experimental group vs. control group 1 and experimental group vs. control group 2). The sample was reactive for all women from the experimental group, 16% of women from control group 1, and one-fourth of women from control group 2, as presented in Table 2.

Age was not significantly correlated with the antibody level, as seen in Table 3. The patients with LSIL diagnosis and those with HSIL or cancer diagnosis did not vary considerably in terms of antibody level and antibody level divided by cut-off value, as shown in Table 4.

Samples were reactive for 93% of women who received an LSIL diagnosis and 87% of women who received either an HSIL or cancer diagnosis.

Comparison made with Student’s *t* test.

## 4. Discussion

Our study aimed to assess the level of anti-HPV antibodies in patients with diagnosed cervical pathology and in healthy patients after vaccination. Our work supports the practice of vaccinating HPV-infected patients after sexual initiation by showing the level of antibodies persisting after vaccination. Despite the proven and indisputable effectiveness of the 9-valent HPV vaccine as primary prevention in juveniles before sexual initiation, its efficacy has not yet been demonstrated in women with diagnosed cervical pathology.

As expected, in all patients vaccinated with the 9-valent vaccine, the samples turned out to be reactive. Additionally, the analysis confirmed a relationship between levels of antibodies and vaccination status. We noted significantly higher levels in vaccinated patients than in those of the unvaccinated control groups: 1.77 vs. 0.09 and 0.13, respectively. These results are consistent with the work published by Mirte Scherpenisse et al. Naturally induced HPV-specific antibodies from single-positive sera were genotype-specific and neutralising.

In contrast, the antibodies of multi-positive sera were less genotype-specific, cross-reactive, and tended to be non-neutralising. Post-vaccination antibody avidity was approximately three times higher than after HPV infection [16]. Post-vaccination antibody status assessment may help analyse the effectiveness of HPV preventative vaccination programmes. Vaccine efficacy against HPV16 and 18 infections were sustained over eight years post-vaccination [17]. HPV-specific IgG antibody levels and its neutralising activity remained well above the antibody levels induced by HPV infection [17,18]. Additionally, HPV vaccines offer cross-protection against several non-vaccine HPV types in patients without a previous HPV infection [19]. Antibodies that were also capable of neutralising non-vaccine HPV types were most frequently found to be directed against HPV31 and 45. Cross-neutralising antibody levels against HPV31, 33, 35, and 45 were significantly associated with their phylogenetically related vaccine-type antibody levels [20]. HPV genotypes frequently detected in cervical cancer are as follows: 31, 33, 45, and efficacious vaccines against these HPV types might further reduce malignancies. However, vaccine efficacy against non-vaccine HPV types decreased rapidly over time [19]. 

Interestingly, our results indicate that a current or persistent infection with human papillomavirus gives a lower antibody level percentage than in vaccinated patients, which is consistent with the reports presented by other researchers. In our study group, 18% of women infected with HPV had clinically significant levels of L1-HPV antibodies, and for comparison, we observed the antibodies in 100% of vaccinated women. Investigators argue that the rate of seroconversion associated with the vaccines is high, namely, >99% in women and men [21,22,23,24]. In contrast, the seroconversion results from natural infection are an estimated 50–70% in women [25] and men [26].

Although most of the patients in the control groups had antibody levels below the cut-off, we observed 3/20 reactive samples in control group 1 and 2/8 in control group 2. It is worth noting that only one woman with a reactive sample had no burden. In other cases, we observed the presence of infections with HR HPV and histopathologically confirmed HSIL or a history of Hashimoto’s disease. This observation may provide new insight into the factors modulating the immune system. It is possible that comorbidities, dysfunctions of the immune system, or infection with HPV genotype 16 strongly stimulate the immune system to produce antibodies. Data provided by Aubin et al. suggest that autoimmune inflammatory diseases (AIID) and the drugs used to treat them are associated with an excess risk of genital HPV infection. Although this excess risk has not been specifically evaluated, the available data indicate a need for close monitoring of patients with AIID, regardless of their treatment, to ensure the prevention and treatment of benign and premalignant lesions [27]. 

Petter et al. [28] indicated that serological assays for HPV could help identify patients at risk of HPV-related cancers. In addition to strategies connected with antibody detection, DNA sequencing or the PCR method are also widely used to detect the viral DNA of HPV in tissue samples [29]. Therefore, antibody- and DNA- based assays can complement each other for the reliable identification of HPV-infected patients. 

Researchers from Mexico presented somewhat similar work. Their study aimed to assess type-specific cervical HPV prevalence and their association with HPV-specific antibodies in a cohort of female university students. The observed study group was similar in terms of number. HPV genotyping was performed by amplifying and sequencing a fragment of the L1 protein. In addition to sexual behaviour, it was observed that the presence of serum-specific IgG antibodies against HPV can impact the prevalence of the virus. Alexander Pedroza-Gonzalez et al. suggest that seropositivity to HPV-16 and HPV-18 was associated with a lower prevalence of HPV-16, but not for other HPV types. Of note, there was a lower proportion of HPV-specific seropositivity in women who had the presence of the same HPV type in a cervical specimen, suggesting an immunoregulatory mechanism associated with the viral infection [30].

Efforts towards the detection of HPV antibodies as a tool to monitor and assess vaccine efficacy have increased significantly in recent years. In the study conducted by Bhatia et al., a standardised ELISA test developed for anti-HPV16L1 antibodies was validated against the WHO’s international positive serum standard for HPV16. This assay was amenable to both venous blood and dried blood spots. The researchers also admit that the sample size used for the study was small; however, the presented technique has promise for widespread use in epidemiological and field studies of antibody prevalence and, coupled with the avidity measurement, may be of use in individual cases for monitoring vaccine responses such as failures [14].

A relatively small research group limited our methodological choices. However, in the future, we will be able to expand the group and test the level of antibodies over the next few years to assess the trend of changes. Fortunately, we observe an increasing awareness of patients and their partners and a growing number of vaccinations against HPV in both adults and those at the pre-contraceptive age.

## 5. Conclusions

The results of our study may indicate that high levels of antibodies are maintained after HPV vaccination. This further suggests that vaccinations are also effective in subjects after sexual initiation. These conclusions might help identify patients with immune disorders who have survived the infection with HR HPV, resulting in changes in the intraepithelial neoplasia. Analysing specific immune disorders might help identify groups of women with the highest risk of developing HSIL (CIN 2, CIN 3) lesions and, consequently, malignancies. 

## Figures and Tables

**Figure 1 vaccines-10-00728-f001:**
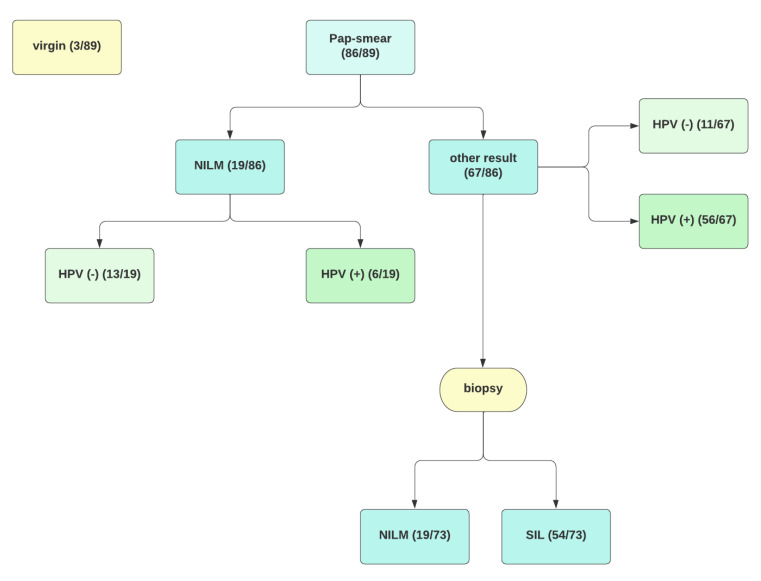
Flow chart. SIL—squamous intraepithelial lesion, NILM—negative for intraepithelial lesion or malignancy, HPV—human papillomavirus.

**Figure 2 vaccines-10-00728-f002:**
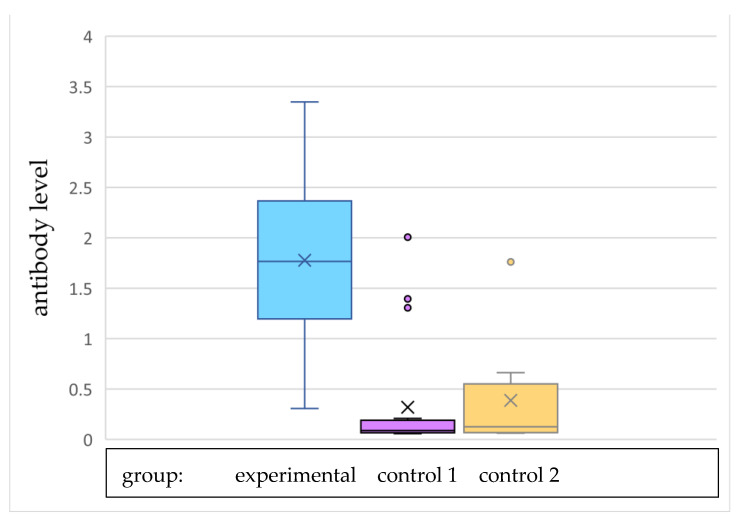
Antibody level.

**Figure 3 vaccines-10-00728-f003:**
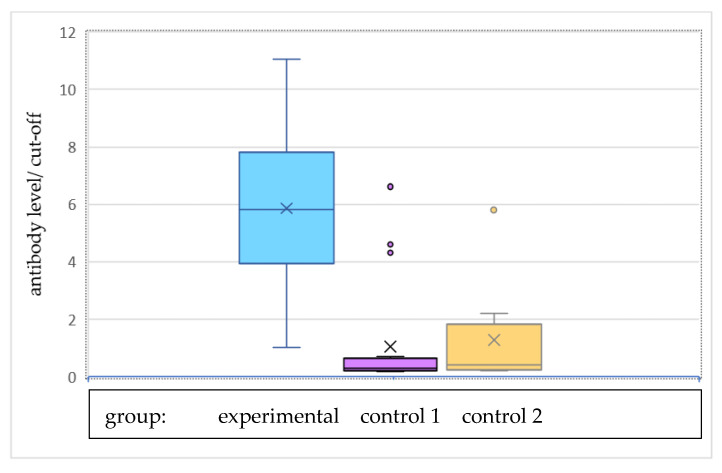
Antibody level divided by the cut-off value.

**Table 1 vaccines-10-00728-t001:** Basic characteristics of study and control groups.

Group	*n*	HPV Status	Biopsy Result	Vaccination Status
Experimental	61	(+)/(−)	Normal/LSIL/HSIL	+
Control 1	20	(−)	Excluded LSIL and HSIL	−
Control 2	8	(+)/(−)	LSIL/HSIL	−

HPV—human papillomavirus, *n*—number.

**Table 2 vaccines-10-00728-t002:** Detailed group characteristics.

Characteristic	Experimental Group	Control Group 1	Control Group 2	*p*1	*p*2
*n* = 89	61	20	8		
Age, M ± SD	34.03 ± 7.32	36.40 ± 7.59	32.88± 7.77	0.217 ^1^	0.667 ^1^
Number of term pregnancies, *n* (%)					
0	26 (42.6)	7 (35.0)	5 (62.5)	0.873	0.249
1	18 (29.5)	8 (40.0)	0 (0.0)		
2	14 (23.0)	4 (20.0)	3 (37.5)		
3	3 (4.9)	1 (5.0)	0 (0.0)		
Number of pre-term pregnancies, *n* (%)					
0	60 (98.4)	20 (100.0)	8 (100.0)	>0.999	>0.999
1	1 (1.6)	0 (0.0)	0 (0.0)		
Number of miscarriages, *n* (%)					
0	55 (91.2)	18 (90.0)	8 (100.0)	0.797	>0.999
1	4 (6.6)	2 (10.0)	0 (0.0)		
2	2 (3.2)	0 (0.0)	0 (0.0)		
Number of pregnancies, Me (Q1; Q3)	1.00 (0.00; 2.00)	1.00 (0.00; 2.00)	0.00 (0.00; 2.00)	0.991 ^2^	0.421 ^2^
Comorbidities, *n* (%)	22 (37.7)	6 (30.0)	1 (12.5)	0.600	0.246
Cytology, *n* (%)	61	20	8		
NILM	7 (11.5)	12 (60.0)	0 (0.0)	0.001	0.903
ASCUS	12 (19.7)	5 (25.0)	2 (25.0)		
ASC-H	10 (16.4)	1 (5.0)	1 (12.5)		
LSIL	20 (32.8)	1 (5.0)	3 (37.5)		
HSIL	6 (9.8)	1 (5.0)	2 (25.0)		
AGC	2 (3.3)	0 (0.0)	0 (0.0)		
Virgin	3 (4.9)	0 (0.0)	0 (0.0)		
Cervical cancer	1 (1.6)	0 (0.0)	0 (0.0)		
HPV, *n* (%)	61	20	8		
Positive	52 (85.2)	2 (10.0)	8 (100.0)	<0.001	>0.999
Negative	6 (9.9)	18 (90.0)	0 (0.0)		
Virgin	3 (4.9)	0 (0.0)	0 (0.0)		
Biopsy, *n* (%)	*n* = 57	*n* = 8	*n* = 8		
NILM	11 (19.3)	8 (100.0)	0 (0.0)	<0.001	0.285
LSIL (CIN 1)	17 (29.8)	0 (0.0)	1 (12.5)		
HSIL (CIN2, CIN 3)	28 (49.1)	0 (0.0)	7 (87.5)		
Adenocarcinoma	1 (1.8)	0 (0.0)	0 (0.0)		
Antibody level, Me (Q1; Q3)	1.77 (1.22; 2.35)	0.09 (0.07; 0.19)	0.13 (0.07; 0.44)	<0.001 ^2^	<0.001 ^2^
Antibody level/cut-off, Me (Q1; Q3)	5.83 (4.01; 7.77)	0.29 (0.22; 0.62)	0.41 (0.23; 1.45)	<0.001 ^2^	<0.001 ^2^
Reactive, *n* (%)	61 (100.0)	3 (15.5)	2 (25.0)	<0.001	<0.001

NILM—negative for intraepithelial lesion or malignancy; ASCUS—atypical squamous cells of undetermined significance; ASC-H—atypical squamous cells cannot exclude HSIL; LSIL—low-grade squamous intraepithelial lesion; HSIL—high-grade squamous intraepithelial lesion; AGC—atypical glandular cells; Q1—first quartile; Q3—third quartile; *n*—number, *p*1—*p*-value for comparison between experimental group and control group 1; *p*2—*p*-value for comparison between the experimental group and control group 2. Comparisons were made with Student’s *t* test ^1^ or Mann–Whitney U test ^2^ for quantitative variables and Fisher’s exact test for qualitative variables.

**Table 3 vaccines-10-00728-t003:** Correlation between age and antibody level.

Variable	Age	
	r	*p*
Antibody level	−0.11	0.137

r—Pearson’s r correlation coefficient. *p*—*p*-value.

**Table 4 vaccines-10-00728-t004:** Comparison of antibody level and antibody level/cut-off between groups with different diagnoses.

Variables	Diagnosis		*p*
	LSIL (CIN 1) *n* = 18	HSIL (CIN 2, CIN 3) and cancer *n* = 36	
Antibody level (M ± SD)	1.63 ± 0.88	1.53 ± 0.89	0.691
Antibody level/cut-off (M ± SD)	5.39 ± 2.90	5.04 ± 2.92	0.691

*p*—*p*-value; LSIL—low-grade squamous intraepithelial lesion; HSIL—high-grade squamous intraepithelial lesion; *n*—number.

## Data Availability

Not applicable.

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
