# Peer review of "Immunity after HPV Vaccination in Patients after Sexual Initiation"

_vaccines, 2022, doi:10.3390/vaccines10050728_

Round 1

Reviewer 1 Report

Thank you for the opportunity to review this manuscript. Some minor comments are outlined below

Introduction

Line 41 – When do you see IgM with HPV onc infection and how long does it last? When do you see IgG with HPV onc infection and how long does it last? Same questions for vaccination.

Methods

When were the serum for antibodies collected in relationship to last vaccination dose?

Results.

Line 125 – would add (Table 1)

Line 127 – 2 of the 3 were described as percentages but 1 of the 3 was described as a number. Would use the same convention (preferably percentage) for all 3.

Table 1- need to add number for the lines of the control group 1 and control group 2. Missing under cytology and HPV

Table 3 – is the LSIL and HSIL a cytologic or histologic or combination diagnosis. Need to be clear in the table or results.

Larger numbers in the control groups would strengthen this submission

Reviewer 2 Report

The manuscript entitled “assessment of the level of L1 HPV serum antibodies after 9-valent vaccination” by Pruski et al. is an interesting study on L1 HPV serum antibodies. The author conducted a detailed study to address the major challenge in this area. The manuscript is recommended for publication in the Vaccine journal. However, the author needs to address the following comments before accepted

Comments to author

  1. The abstract needs to be improved
  2. Disclose the abbreviation at their first appearance in the manuscript.
  3. The author should disclose the criteria in the selection of patients and age, gender details if ethically allowed to share in the manuscript
  4. Highly recommended to include a scheme for the study overview
  5. Figure 1 and Figure 2 quality need to be improved
  6. Font size in Figures needs to be increased to readable size
  7. In Figures 1 and 2 there are data points presented as small dots. Suggested to increase the size to make it easy to read
  8. The conclusion also needs to improve significantly and discussed the major outcome of the study
  9. Suggested to the author to cite the literature https://doi.org/10.1016/j.jddst.2022.103351
  10. Except for four references from 2017 and 2021, all are before 2014 literature cited. The author should look at recent progress in this area and cite/replace the recent literature as many as possible
  11. If any similar clinical studies are reported in the literature, please compare this study and discuss how this study is better/advantages/benefits over other
  12. The title of the manuscript does not clearly giving an idea about the study. So, try to make it more suitable title

Round 2

Reviewer 2 Report

Author addressed the comments and hence recommended for publications